# Cancer Associated Endogenous Retroviruses: Ideal Immune Targets for Adenovirus-Based Immunotherapy

**DOI:** 10.3390/ijms21144843

**Published:** 2020-07-08

**Authors:** Amaia Vergara Bermejo, Emeline Ragonnaud, Joana Daradoumis, Peter Holst

**Affiliations:** 1InProTher, Bioinnovation Institute, Copenhagen Bio Science Park, 2200 Copenhagen, Denmark; era@inprother.com (E.R.); jdb@inprother.com (J.D.); 2Center for Medical Parasitology, Department of Immunology and Microbiology, University of Copenhagen, 2200 Copenhagen, Denmark

**Keywords:** adenovirus, endogenous retrovirus, immunotherapy, cancer, virus-like-particles, virus-like-vaccines, immunology, adenoviral vector, envelope

## Abstract

Cancer is a major challenge in our societies, according to the World Health Organization (WHO) about 1/6 deaths were cancer related in 2018 and it is considered the second leading cause of death globally. Immunotherapies have changed the paradigm of oncologic treatment for several cancers where the field had fallen short in providing competent therapies. Despite the improvement, broadly acting and highly effective therapies capable of eliminating or preventing human cancers with insufficient mutated antigens are still missing. Adenoviral vector-based vaccines are a successful tool in the treatment of various diseases including cancer; however, their success has been limited. In this review we discuss the potential of adenovirus as therapeutic tools and the current developments to use them against cancer. More specifically, we examine how to use them to target endogenous retroviruses (ERVs). ERVs, comprising 8% of the human genome, have been detected in several cancers, while they remain silent in healthy tissues. Their low immunogenicity together with their immunosuppressive capacity aid cancer to escape immunosurveillance. In that regard, virus-like-vaccine (VLV) technology, combining adenoviral vectors and virus-like-particles (VLPs), can be ideal to target ERVs and elicit B-cell responses, as well as CD8^+^ and CD4^+^ T-cells responses.

## 1. Introduction

Adenoviral vectors have been used for decades in experimental vaccines against retroviruses, such as the human immunodeficiency virus (HIV), and the most powerful immunisation schedules tried in humans are adenovirus based prime-boost regimens [1]. The extensive use of adenoviral vectors to transfer and deliver genes is partly explained by their relatively large coding capacity. This has been one of the major focuses in recent studies, where HIV retrovirus-based virus-like-particles (VLPs) have been encoded in adenoviral vectors, consolidating the concept of virus-like-vaccines (VLVs). This innovative technology consists of an adenoviral vector encoding the group specific antigen (GAG) upstream of a viral envelope (ENV) protein linked by a self-cleavable peptide (P2A). GAG is sufficient for VLP release and the retroviral *ENV* gene encodes sufficient information in the cytoplasmic tail for appropriate antigen display. The result is that adenovirus transduced cells become in situ producers of VLPs. Immunologically this is a perfect scenario: direct transduction provided by adenoviral vectors yields optimal CD8^+^ T-cell (cytotoxic T-cells) responses, and secreted VLPs present structurally accurate antigen providing optimal CD4^+^ T-cell (helper T-cells) and antibody producing B-cell responses) [2].

While cancer remains a major challenging disease in our societies [3], it has been natural to use knowledge from the vaccine field to target it. Recent evidence in the field of cancer immunotherapy has conclusively linked the interaction of tumour antigen specific CD4^+^ and CD8^+^ T-cells in the tumour microenvironment with effective tumour rejection [4]. It has also been shown that the formation of B-cell containing secondary lymphoid structures is necessary to maintain T-cell functionality [5]. Hence, such adenovirus based VLVs would make, theoretically, ideal anticancer therapies by eliciting all components of a successful antitumor response.

Stimulating the right immunity, however, only matters if the relevant antigens are expressed by the cancer cells and are accessible to the immune system. In that regard, using endogenous retroviruses (ERVs) as targets seems to be the immediate solution but, accordingly, it requires that immunogenic forms of retroviruses are also expressed in cancers. Fortunately, finding retroviruses in cancers is not problematic as ERVs constitute 8% of the human genome [6]. Nevertheless, it has been a subject of debate, which, if any, of the specific ERV types can be found with sufficient cancer specificity to be targeted by immunogenic adenovirus based VLVs.

Here, we review the evidence of ERVs as ideal targets for adenovirus-based immunotherapy starting with an overview of the mouse models, where a definitive ERV tumour-promoting role is strongly suggested. This is followed by the human data where, likewise, a strong ERV involvement can be found in cancer, at least in cell-based assays. Additionally, we describe some genetic studies indicating ERV families as targets that could be used for therapies against specific cancer groups. Succeeding the discussion on the ERV involvement in cancer, we discuss the previous attempts at targeting ERVs in cancer. Lastly, we highlight the experiences derived from using adenovirus vectors in anti-HIV vaccines and in immunotherapeutic studies to conclude with the emerging experience of using them to target ERVs—a novel strategy in cancer immunotherapy.

## 2. Endogenous Retrovirus in Cancer

### 2.1. Lessons from Rodents

Unlike humans, rodents have ERVs of which infectious counterparts still exist. Therefore, the necessary reagents for studying rodent ERVs and detecting an association with cancer became available rather early, suggesting a near uniform presence of C-type retroviruses in carcinogenesis [7]. An early suggestion of a causal role of ERVs in tumour development was provided by Whitmire et al. when C-type RNA viral vaccines were able to reduce chemically induced sarcoma tumour development in mice [8]. Similar findings were obtained a decade later in rats injected with endogenous rat retroviruses. Additionally, animals showed a significant protection against the development of induced fibrosarcoma [9]. The murine ERV molecule responsible for the tumour promoting effects was strongly suggested to be the *Env* gene, as its overexpression was found to allow engraftment of cells normally incapable of growth in immunocompetent mice. Moreover, part of the effect was shown to be local in the growing tumours [10]. The *Env* gene was later targeted by RNA interference in Mangeney et al., where the authors demonstrated the importance of *Env* since knocking down the murine ERV led to tumour rejection in wild type (WT) mice. The knock down allowed murine melanoma tumour cells to be recognised by the immune system and tumour growth to be arrested, which led to an increase in survival. These beneficial effects depended on a functional immune system, suggesting that the mouse ERV acted as an immune modulator. To identify the molecular target, the expression of ENV protein was restored by overexpression, leading to reversion of tumour rejection. Interestingly, and highlighting an immune mediated effect, injecting T regulatory (T_reg_) cells from a control melanoma-engrafted mouse allowed tumour cells to grow, despite the absence of murine ERV gene expression [11].

The molecular studies with *Env* overexpression and inhibition clearly pointed to a causal involvement in at least some murine cancers. Quite staggeringly, the cancer types that had trouble developing when ERVs were targeted include carcinogen-induced cancers, which suggests a rather broad involvement. Indeed, Scrimieri et al. demonstrated a broad association of high levels of retroviral *ENV* expression with a panel of different tumour cell lines, and absence of *ENV* expression in normal tissue. Interestingly, a few tumour cell lines only express low levels of endogenous retrovirus and a closer look at these types revealed that most of them express human papilloma virus (HPV) oncogenes [12].

### 2.2. Human Endogenous Retrovirus

The search for human ERVs (HERVs) started as genetic tools became available. Reports in the last few decades suggest a wide association of HERVs with many cancer types and few of these viruses are broadly or consistently associated with specific cancer types (Table 1). Undoubtedly, the most broadly and consistently cancer associated ERV type is HERV-K, followed by HERV-H and HERV-W/syncytin 1 followed by HERV-R.

### 2.3. HERV-K

Among the different HERV-K subfamilies, the most studied is called HML-2 [13]. This sequence has been described as the most recently active and well preserved with regards to new integrations in the human genome. It is part of a bigger group englobed by the HERV-K family and it has been implicated in several diseases [14]. Among the cancer types linked to HERV-K, breast cancer stands apart in the volume of research and its extensive validation for causal involvement. 

Wang-Johanning et al. have been able to show in several publications that both full length proviral and spliced *ENV* transcripts are present in most breast cancer tissues and that the expression level is negatively associated with survival [15,16]. Her research team has also exposed that these transcripts have coding capacity and that the expression of ENV protein is significantly higher in breast cancer cells than in normal epithelial cells. Within a cohort of breast cancer patient serum samples, they were able to demonstrate the presence of the surface unit (SU) of HERV-K ENV protein and antibodies against HERV-K ENV. They also used autologous dendritic cells (DCs) pulsed with HERV-K *ENV* SU RNA to successfully detect HERV-K specific T-cell responses in the patients’ PBMCs and tumour infiltrating lymphocytes. Lastly, the team was able to distinguish basal carcinoma as a subgroup of cancers with the most pronounced increased expression of HERV-K *ENV* from a cohort of 512 breast cancer patients with invasive ductal carcinoma [16,17,18,19,20].

In addition to studies associating the presence of HERV-K and cancer prognosis in patient samples, in vitro and in vivo studies have backed up this correlation. In breast cancer cells, interference of *ENV* expression by shRNA reduced proliferation, migration and invasion, resulting in smaller tumours when they were engrafted in immunodeficient nude mice. Importantly, the shRNA targeting could be seen to reduce expression of Ras, p-RSK and p-ERK by single cell RNA seq, which are all part of oncogenic pathways [15]. Conversely, overexpressing HERV-K *ENV* resulted in activation of the ERK pathway and epithelial to mesenchymal transition (EMT), leading to an invasive and migration profile. At the molecular level, the signalling events were linked to the cytoplasmic domain of ENV [21].

In addition to breast cancer, pancreatic, ovarian and prostate cancers have also demonstrated a functional role of HERV-K. Feng Wang-Johanning’s team has used shRNA interference to study pancreatic cancer progression achieving similar and significant results to what was observed in breast cancers. This includes diminished cell growth and impairment of RAS, p-ERK, p-RSK and p-AKT expression following HERV-K knock-down [15]. In addition, in a study by Li et al., diminished metastasis rate was also observed by downregulating HERV-K [22].

We can find the same pattern in ovarian cancers. As mentioned for breast cancer, pulsing DCs with HERV-K ENV antigens led to T-cell responses (measured by T-cell proliferation and INFγ production) much stronger in ovarian tumour patients than in normal or benign patient samples. In a cohort of 89 patients with both ovarian cancer and benign ovarian diseases, cancer patients showed increased mRNA levels of HERV-K, HERV-R and HERV-E when compared to healthy controls. Additionally, HERV-K ENV protein expression on the surface of the patients’ tumours was also significantly higher than that in immortalised ovarian non-tumorigenic cells when assessed with both flow cytometry and immunohistochemistry. Furthermore, the HERV-K ENV protein and the reverse transcriptase were also detected in the plasma of the patients by Western blot [23,24,25]. Apart from the mentioned HERVs, there is data suggesting that the ERV HEMO also has an increased expression in ovarian tumours when compared to normal tissue [26].

Regarding prostate cancer, both full-length and spliced forms of HERV-K were detected in the prostate cancer cell lines LNCaP, DU145, PC3 and VCaP [27]. Several studies show increased expression of HERV-K in men with prostate cancer compared to healthy controls. However, it was observed that only certain HERV-K loci, such as HML-2 H22q and ERVK-5, were activated and overexpressed in patients. This rise was also associated with higher plasma IFN-γ levels. A subset of patients, showed an increase expression of HERV-K HML-2 H22q in their tissues, which was linked to GAG protein expression. Their serum also revealed that 6.8% had antibodies against HERV-K GAG protein (1.8% in healthy patients), and that they were more frequent in advanced than in early prostate cancer (21% to 1.4%). HERV-K GAG protein could also be detected using prostate cancer autologous serum. Additionally, immunohistochemistry of prostate tumours showed a significant upregulation in the expression of the HERV-K ENV protein [28,29,30,31].

Other cancers have also been related to HERV-K, like renal carcinoma [32,33], lymphoma [34], leukaemia [35], melanoma [36,37], sarcoma [38], endometrial [39] or lung cancer [40]. However, we chose to focus on the more extensively researched ones.

### 2.4. HERV-H and HERV-W

As mentioned above, several ERVs have been reported to be expressed in different tumours, and even if HERV-K is the most extensively researched, other ERVs may also have a role in oncogenesis, especially HERV-H and HERV-W.

In the case of HERV-H, it has been shown to be expressed in colon cancer. Even if there are several different loci of HERV-H, all of them are generally more actively transcribed in cancer tissues than in the adjacent normal tissue [41]. The mechanism regulating the expression of HERV-H has been linked to hypomethylation, which has also been shown to lead to increased expression of HERV-K, HERV-W and HERV-P in some colon cancer cells lines [42].

Apart from colon cancer, HERV-H has been associated with a potential role in head and neck squamous cell carcinoma (HNSCC) development [43]. HNSCCs are considered highly immune suppressive and they stand out among the investigated human cancers given that the global association between ERV expression and cytolytic T-cell activity is negative [44]. More recent papers have also been able to link HNSCC with HERVs. For example, Michna et al. have suggested HERV-R to have a role in radiation resistant tumours [45].

Transitional cell carcinoma (TCC), also known as urothelial cell carcinoma (UCC) of the bladder, was also linked to ERV expression. HERV-W *ENV* mRNA levels and protein expression in UCC patient tissue was higher (75.6%) when compared to tumour-adjacent tissue (6.1%). Similar levels were found when staining the tissue by immunohistochemistry (78% to 7.3%). In vivo experiments have additionally demonstrated the oncogenic coupling of HERV-W, as it was able to induce proliferation and tumour formation [46]. HERV-E expression was also found to be high in a number of UCC patients and proven to interfere with a phospholipase which could lead to tumorigenesis [47]. Finally, trough immunohistochemistry; Lin et al. recently demonstrated an increase expression of the HERV-H long terminal repeat-associating protein 2 in bladder urothelial carcinoma compared to normal bladder tissue. This HERV-H protein expression proved to be associated to tumour size, stage, grade and lymph node metastasis [48].

In addition to the aforementioned relevant examples, there is a broad registry of ERVs in other cancers. HERV-R has been detected in renal carcinoma [49], leukaemia [50] and lung cancer [40]. HEMO has been related with lung cancer and renal carcinoma [26]. HERV-W’s active presence has been detected in lymphoma [51,52], endometrial carcinomas [39] and in seminomas [53].

Nonetheless, some of the HERVs are also expressed in normal tissues, usually to a lower extent that in the cancer associated tissue. For example, HERV-K transcripts have been found by RT-qPCR in few uninvolved breast tissues, stomach and small intestine, but with a significantly lower expression than the tumour tissue, and HERV-H and HERV-E have also been found in normal pancreatic tissue [54,55,56]. There are also HERVs that are broadly involved in non-tumorigenic processes such as pregnancy. HERV-K, HERV-R and HEMO have been found in placenta. The *ENV* region of the latter was found in pregnant women’s blood [17,26,57]. The ENV proteins of HERV-W and HERV-FDR, also known as syncytin 1 and syncytin 2, have been widely described to play a role in the immune recognition of placenta. The immunosuppressive domains of these two proteins are able to activate the MAP kinase pathway modulating Th1 response, eventually facilitating foetal development [17,58,59,60].

Taken as a whole, the studies of human ERVs have made a comeback and now point to equally strong cancer involvement as seen in the pivotal and conclusive rodent studies from the last decades. Likewise, evidence suggests a degree of cancer selectivity which might make ERVs good cancer biomarkers and desirable targets for immunotherapies.

## 3. Endogenous Retrovirus and Immune Evasion

Based on the resemblance between ERVs and HIV, researchers have tried to elucidate if there is a structural resemblance that allows the immune evasion. HIV research established the presence of an immunosuppressive domain in the transmembrane subunit of the ENV protein which regulates gene expression and cytokine release [61]. The early records of the role of ERVs in immune evasion already pointed to the ENV protein for the failure of the effector cells to detect the target, which allowed tumours to avoid immune recognition [62]. More specifically, the immunosuppressive effect was linked to the transmembrane (TM) region of ENV (called p15E) [63]. The p15E region contains what has been described as an immunosuppressive domain (ISD) (Figure 1). The respective ISD in HERV-K was shown to inhibit activation of PBMCs and modulate cytokines as well as gene expression, suggesting a strong connection between this TM region and immunosuppressive activity. Additionally, HERV-K positive particles released from a human teratocarcinoma cell line were shown to modulate cytokine release [64]. Likewise, HERV-H’s ISD was described to have the capacity to regulate immune escape [65]. Furthermore, decades old research unequivocally demonstrates that immunosuppressive peptides present in the serum of HNSCC patients cross-react with antibodies against mouse gamma retrovirus, like the murine antibodies against p15E [66,67,68]. Monoclonal antibodies against the immunosuppressive domain of the p15E protein demonstrated the important role of this region in the *ENV* protein as p15E depleted serum lost the immunosuppressive activity [69].

The importance of the immunosuppressive function of the TM domain has been further confirmed for several other ERVs, such as HERV-W and HERV-FDR, in relation with foetal development, where immunosuppression was defined by the ability of these proteins to inhibit the stimulated expansion of PBMCs and T-cells [70]. The ISD domain of syncytin-2 (ENV protein of HERV-FDR) was found to induce ERK1/2 phosphorylation and inhibit TNF-α production, inducing activation of PBMCs and cytokine production. Interestingly, the different structures that the ENV protein can take, might be the key for the inhibitory potential, as it has been shown that monomer forms of the ISD are not inhibitory, while dimers or trimers are. Furthermore, the ability of the immune suppression to work in a macromolecular context was confirmed when syncytin-2 containing exosomes were linked to especially strong inhibitory capacity against PBMCs [60].

Exosomes (VLPs) containing other ERVs have been studied too. For the first-time, zebrafish embryos were injected with HERV-K positive VLPs generated from two colon cancer cell lines. Expression of the pro-inflammatory cytokine IL1-β was lower in cancer derived extracellular vesicle injected samples, compared to control samples. Additionally, there was a strong association between cytokine release and HERV-K presence, indicating a role in immunomodulation of the immune response and an effect in tumour progression [42].

Overall, it is quite clear that cancer cells manage to avoid recognition by the immune system with the help of the ISD present in the ENV protein encoded by ERV sequences. The extensive association of tumorigenesis and the presence of ERVs, in combination with their role in immune evasion, make ERVs an interesting target candidate for immunotherapy against cancer.

## 4. Targeting of Endogenous Retrovirus in Cancer

As mentioned before, already in the late 1900s some cancers responded to treatment targeting murine ERVs, with perhaps the most relevant one being Thiel et al. Their team slowed the development of progressed leukaemia by a combination therapy with specific antibodies targeting both ERVs’ TM and SU ENV proteins (with limited effect of monotherapy) [71]. Moving further, several studies substantiated significant effect with T-cell vaccinations using the ENV protein of the murine ERV against cancer. DNA vaccination encoding *ENV* exhibited a high efficacy against tumours eliciting antigen specific CD8^+^ T-cells. Therefore, vaccination efficacy has been improved boosting the CD8^+^ T-cell activity by using adjuvants such as CD40 or co-expressing other elements such as with β-galactosidase or part of a tetanus toxin sequence [72,73,74].

Wang-Johanning et al. suggested in 2006 already that HERV-K proteins could act as tumour-associated antigens (TAAs) and used as targets in immunotherapy [24]. Besides that, their group showed the potential of a monoclonal antibody (mAb) against HERV-K ENV inhibiting breast cancer xenograft tumours in vitro [75]. Moreover, knocking down HERV-K decreased the expression of *ENV* and inhibited pancreatic cell proliferation and transformation in vitro. In mouse xenograft models challenged with knocked down HERV-K pancreatic tumour cells, the tumour growth was reduced with absence of lung metastasis [22]. As mentioned before, Wang-Johanning et al. also managed to generate T-cell responses using DCs pulsed with HERV-K ENV antigens, where these T-cells were able to eradicate HERV-K expressing ovarian cancer cells [23]. Similar results have been shown for lung cancer, where the injection of a modified vaccinia Ankara (MVA) encoding HERV-K *ENV* and *GAG* prevented lung tumour outgrowth and metastasis in mice. The same vaccine actually proved to have prophylactic potential when vaccinated mice did not develop lung metastasis [76,77]

Recently, a patent filing for a peptide-based therapy was published. Three peptides from the HERV-K *GAG* sequence were shown to be able to expand HERV-K specific T-cells ex-vivo, and such T-cells could subsequently eradicate HERV-K expressing tumours (PCT/EP2019/073883).

The last revolution in the cancer immunotherapy field has been the introduction of chimeric antigen receptor (CAR) T-cell therapy, where the immune cells from the patient are re-educated to fight cancer by introducing the sequence of interest into the T-cell receptor (TCR). Zhou et al. have managed to introduce HERV-K specific CAR T-cells by designing a TCR with an anti-HERV-K mAb sequence. The HERV-K CAR T-cells were able to inhibit breast tumour growth in xenograft mouse models and exhibit specific cytotoxicity (enhanced IFN-g, TNF-Į and IL-2) against cancer cells while normal breast cells were not targeted. Adding to previous prophylactic observations, these HERV-K specific CAR T-cells were also able to prevent metastasis to other organs [78]. Using the same strategy, metastatic melanoma mouse models successfully decreased tumour burden after treatment with HERV-K specific CAR T-cells [79].

Targeting of ERVs in cancer is still in its inception with the majority of reports focused on murine ERVs or HERV-K. Notably, previous attempts have been rather simple vaccine designs inducing a single CD8^+^ T-cell epitope or tested in immunologically simplified model systems (e.g., HERV-K transfected cells or xerographs).

## 5. Adenoviral Vectors Used as Anti-Retroviral Vaccines

Adenoviruses (Ads) are non-enveloped viruses containing double-strained DNA genomes ranging from 34 to 43 kb. The selection of adenoviruses as vectors for vaccination strategies is based on their inherent adjuvant characteristics, which trigger innate immunity [80]. Adenoviral vectors, as compared to others such as lentiviruses, retroviruses and adeno-associated vectors, are valuable for their capacity to infect a wide range of non-replicative and replicative cells, including DCs, and being highly immunogenic. The advantage of using adenoviral vectors is that they elicit mild early innate responses and regulate adaptive immune responses against the vector. Concurrently, they avoid transgene expression impairment, promoting strong and broad immune responses against the latter. They are also considered safe due to the elimination of E1 and E3 genes which makes them replicative-defective vectors and allows the insertion of foreign DNA—first generation of adenovirus [81,82]. Apart from their large cloning capacity, adenoviral vectors are relatively easy to produce in high titters and have shown to be relatively safe to use in humans and other mammalian species. Defective recombinant adenoviruses have already been used as vectors to test different vaccination strategies targeting various diseases such as influenza [83], HIV [84], cancer [83,85] and are approved for human use against Ebola [2,86].

HIV is a valuable example of a challenge in vaccine research and development. Despite many attempts, a viable HIV vaccine is still missing in the market. The difficulty to achieve protection, among other reasons, is the need of a combined antibody and T-cell inducing vaccine for full protection. In order to prevent HIV infection and restrict viremia after infection, both antibody and cellular responses are necessary to block virus transmission and eliminate HIV-infected cells. Additionally, robust and effective memory responses are also required to mount a quick, specific and greater secondary response upon re-encounter with the primary antigen [87].

Even the existing powerful techniques, such as live-attenuated viral vaccines, that mimic many aspects of resolved acute infection (e.g., smallpox and measles), have limitations in terms of safety and efficacy against chronic infections such as HIV. The discovery of the VLP technology has allowed great advances in the field due to its versatility and favourable immunological characteristics. VLPs do not only offer many of the same desirable properties as live-attenuated vaccines, but they also constitute an advantageous platform due to their size, repetitive surface pattern, capacity to generate both innate and adaptive immune response, as well as being a safe and economically profitable system [88]. Upon successful expression, VLPs self-assemble mimicking the original parental virus symmetry. This unique conformation arrangement constitutes VLPs as a powerful pathogen associated structural pattern (PASP) which facilitates cross-linking of B-cell receptors [89]. The repetitive surface structure of the VLPs facilitates their opsonisation and uptake by DCs, which process and present them on major histocompatibility complex class II (MHC-II) molecules to activate CD4^+^ T-cells. Additionally, their particulate structure and size eases an efficient cross-presentation of VLP-derived peptides on major histocompatibility complex class I (MHC-I) molecules and therefore the stimulation of CD8^+^ T-cells. This feature constitutes an additional advantage and therefore it has been broadly used when designing VLP-based vaccines for cancer treatments. Actually, several studies have indicated that VLPs are able to overcome the immunosuppressive state of the tumour microenvironment [90,91] and break self-tolerance to elicit cytotoxic lymphocyte responses crucial for destruction of cancer cells [92,93,94].

Despite the potential of VLPs to generate CD8^+^ T-cell responses, their principal outcome is mainly translated into effective triggering of B- and CD4^+^ T-cell responses. So, when compared to other vaccination strategies, such as viral vector-encoded antigens, they do not seem to have an added effect, but to rather show an inferior VLP incorporation [95]. This leads us to think that one way to obtain a more comprehensive, powerful and effective immune response would be to combine the advantages of VLPs with those of viral vectors by simply encoding VLPs within a replication deficient viral vector. This type of vaccination strategy is the already mentioned VLV technology [3].

VLVs are non-replicative viral vectors which produce VLPs in the cells they transduce. This system uses regular virus vectored vaccines to first allow efficient stimulation of CD8^+^ T-cells via the expression of the antigen intracellularly for MHC-I presentation on the surface of transduced antigen presenting cells (APCs) such as DCs (Figure 2). Then, transduced APCs release VLPs which can be presented to DCs and B-cells to further induce CD4^+^ helper T-cells, CD8^+^ T-cells and antibodies. Therefore, this strategy is highly attractive against challenging pathogens such as HIV as it combines the two most efficient technologies for stimulating both arms of the adaptive immune system. Contrary to VLPs, that require both particle and surface antigen stability and adequate useful product life, VLVs only demand that the vector is stably produced and that the encoded antigens are properly folded when produced in situ. However, the latter does not need much stability, since it does not require any type of storage [3]. This technology was first tested using recombinant canarypox vectors and the MVA vector against HIV/simian immunodeficiency virus (SIV). The recombinant canarypox was used as a prime vaccine in the RV144 trial in Thailand where up to 31% of protection was observed against HIV-1 [96]. While MVA vectors are potent CD4^+^ T-cells and antibody inducers, they are not the most efficient at inducing CD8^+^ T-cells [97]. On the other hand, adenoviral vectors, in addition of eliciting CD4^+^ T-cells and antibodies, they are known to be the most potent inducers of transgene specific CD8^+^ T-cell responses so far, which can be further enhanced using genetic adjuvants and used in a heterologous prime-boost regimen [98,99].

The first-generation adenoviral vectors used as VLVs had a cassette encoding *GAG* upstream of a self-cleavable peptide followed by *ENV* inserted in the genome to produce VLPs. In preclinical studies, this vaccine concept demonstrated potent T-cell and antibody responses against HIV/ SIV [100,101]. To further enhance these immune effects, Andersson et al. showed that encoding homologous *GAG* and *ENV* is highly important. The team could observe that mice vaccinated following a heterologous prime-boost regimen (Ad-MVA) with homologous *GAG* and *ENV* had higher and broader GAG specific CD8^+^ T-cell response. This strategy showed more GAG than ENV specific CD8^+^ T-cell responses and higher ENV binding antibody, compared to expressing different *GAG* and homologous *ENV* [100]. Furthermore, increased GAG to ENV specific CD8^+^ T-cell ratio and diversified T-cell responses were obtained when encoding the invariant chain fused to accessory SIV antigens as a genetic adjuvant in the adenovirus [101]. Additionally, these immune responses were previously associated with partial protection in non-human primates (NHP) which indicates a high chance for success in humans.

## 6. Adenovirus as Therapeutic Cancer Vaccines with ERVs as Targets

The exact mechanisms by which ERVs are activated have not been determined yet, but several studies point to epigenetic modifications. More specifically, ERV activation has been associated with DNA hypomethylation, which had already been widely associated with tumour progression [102,103]. In consequence of this activation several oncogenic pathways seem to get upregulated downstream of ERV gene expression [22]. Most notoriously, the ERK1/2 pathway activation has been detected in HERV-K positive cancers, where this activation led to an EMT phenotype [21,60]. Furthermore, the transcription of ERVs not only activates oncogenic pathways, but it also leads to the generation of exosomes resembling VLPs that can express GAG and ENV [60,104,105]. The presence of the ERV proteins in these particles will elicit innate and adaptive immune responses, being able to generate specific CD8^+^ T-cells among other immune responses [106]. Similarly, the viral proteins processed by the cancer cells would be presented by the MHC-I or -II depending if they are processed through the proteasome or in the endosome. However, these cancer-induced immune responses are not able to control tumour development, probably influenced to some extent by the immunosuppressive state generated by the ISD in the p15E ENV protein (Figure 3). A further activation of the immune system is needed to fight the tumour.

The development of vaccines against cancer has remained very challenging. So far only prophylactic vaccines based on VLPs have shown to successfully prevent the establishment of certain cancers caused by viruses (Hepatitis B and HPV) and are now licenced. However, the efficacy of such vaccines only work before the disease is established and only against specific cancers [107]. Therapeutic vaccines must elicit meaningful immune responses, which must reach the tumour site and evade the immunosuppressive microenvironment to eliminate many cancer cells. Such meaningful immune responses, able to control or eradicate tumour growth, have been characterised by the generation of potent tumour antigen specific CD4^+^, CD8^+^ T-cells and B-cells in the tumour microenvironment [4,5]. While adenoviral vectors are known to induce potent and effective T-cell responses towards the encoded transgene, it appears that cancer requires a vaccine that can induce even stronger cellular immune responses. To do so, different strategies have been developed.

One strategy has been to perform a second immunisation with a different adenoviral vector serotype, or another viral vector, to boost the transgene specific memory T-cell responses elicited from the first immunisation. So far, the most promising combination has been to prime with an adenoviral vector and boost with an MVA vector. This strategy not only increased transgene specific CD8^+^ T-cell responses, but also CD4^+^ T-cell responses, which has been associated with enhance protection against several diseases such as malaria, Ebola and hepatitis C virus [108]. 

Another strategy has been to increase the transgene antigen presentation relative to vector antigens. This was surprisingly achieved by the genetic fusion of the TAA to the C-terminal domain of the MHC-II associated invariant chain (Ii) in the viral vectors [109]. Although this technology was originally designed to MHC-II presentation in adenoviral vectored vaccination, it increased direct MHC-I and -II restricted antigen presentation on the surface of transduced DCs. The result was an increase of TAA specific CD8^+^ T cells and a significant reduction of tumour growth in mice challenged with murine B16.F10 melanomas expressing the TAA: GP^33-41^ from lymphocytic choriomeningitis virus (LCMV), even in LCMV tolerant mice [110].

The T-cell adjuvant effect of the Ii technology in adenovirus vectored vaccines has been observed against several antigens from different pathogens/diseases. For instance, in cynomolgus macaques naturally infected with papillomaviruses, the Ii technology fused to ancestral PV antigens in a prime-boost regimen (Ad-MVA) could raise sufficient T-cell responses against a certain type of PV (MfPV3) and eliminate the infection caused by it [111]. This PV is closely related to HPVs and cause cervical cancer. Generally, the use of this genetic T-cell adjuvant in adenovirus vectored vaccines has shown promising results in pre-clinical studies. So far, it has been evaluated in human clinical trials for its safety and immunogenicity with hepatitis C inserts [112], but variants of it seem to be progressing in human cancer vaccines based on teleost invariant chain (WO2020079234) and shark invariant chain [113] by the biotech companies Nousom Srl. and Vaccitech, respectively. Some pre-clinical cancer strategies, aimed at eradicating tumour growth completely, have focused on further enhancing the T-cell adjuvant effect of Ii in adenovirus vectored vaccines. With the aim of enhancing signal 2 (co-stimulatory signalling) or 3 (cytokines production) from transduced DCs, the co-stimulatory molecule, 4-1BBL and the T-cell growth factor IL-2 were co-encoded in the vector. While co-encoding 4-1BBL did not further enhance TAA specific T-cell responses, IL-2 increased TAA-mediated survival and tumour control in mice, without affecting memory T-cell responses [114,115]. These results suggest the difficulty in further improving the T-cell adjuvant effect of the Ii technology in adenoviral vectored vaccines. One explanation could be that the acutely expanded T-cells negatively impact the longevity of antigen persistence. If this is the case, alternative approaches will be needed to increase the clinical efficacy beyond the invariant chain-based vaccination approach.

Indeed, VLV technology was applied against cancers using an adenoviral vector encoding ERV *ENV* packaged into *GAG* based VLPs. These vaccines were highly effective at managing to target growing colorectal carcinomas and eliminating small tumours in mice. Unlike targeting with invariant chains, using neoantigens where T-cells were the only adaptive mediators of tumour resistance, it was observed that the effect of therapy was dependent on both CD4^+^ and CD8^+^ T-cells combined [116,117]. Following the success in murine ERV, we are currently taking this concept forward targeting human ERVs. To further enhance vaccine efficacy, we are using point mutations in the aforementioned ISD of the ENV protein (WO/2019/043127).

## 7. Discussion

With the high number of people with cancer worldwide and the expectation of this number to increase drastically in the coming years [3], there is a need to develop therapeutic vaccines which can eliminate already established cancers. In comparison to prophylactic vaccines, therapeutic vaccines must circumvent major obstacles like low cancer immunogenicity, an advanced stage of cancer progression and the immunosuppressive microenvironment [118,119,120]. In addition, the choice of the vaccine antigen is very important as it should be specific to the tumour and, if possible, expressed by many different types of cancer for broader coverage. Most work has been focused on tumour-specific antigens or neoantigens that can be presented by the cancer cells to the immune system and elicit tumour-specific immune responses. For instance, ERV proteins are overexpressed on many different types of human cancer and are presented to the immune system, making a relevant and promising target for therapeutic purposes.

Adenoviruses have been used in thousands of reports as vaccines to target cancer. In that role, adenoviruses have a potent cell transduction and stimulate robust CD8^+^ T-cell responses by virtue of direct infection of APCs. Despite these processes, adenoviruses have not been successful in human cancer yet. We speculate that a major drawback of adenoviral vectors is the competition between vector antigens and transgene encoded target antigen for immune dominance, where vector proteins block responses to minor responses in the transgene [121]. Such limitations can be overcome by applying genetic adjuvants, such as the invariant chain. Genetic adjuvants selectively increase transgene specific antigen presentation, as compared to vector antigen presentation, and can be critical to induce cancer specific immunity against self-antigens [110]. It is tempting to apply similar vaccine designs in the targeting of endogenous retrovirus, but to do so will mean missing several opportunities for intervention. Endogenous retroviruses are surface expressed and functional targets on cancers, which mean they can be targeted by B-cells and CD4^+^ T-cells in addition to CD8^+^ T-cells. We believe that the mentioned strategies will fail to exploit the full immunogen capacity of the ERVs [117].

As mentioned before, ERVs can be reactivated and overexpressed in many cancer cell types. Like VLPs, they can trigger both humoral and T-cell responses through the expression of ENV protein on the cell surface, and the secretion of ERV positive exosomes and particles. Although they can certainly trigger immune responses, they are also immunosuppressive. Therefore, a vaccine capable of breaking this immune suppression through the generation of robust T- and B-cell responses is required and adenovirus-vectored VLVs appear to fulfil this requirement (Figure 4).

## Figures and Tables

**Figure 1 ijms-21-04843-f001:**
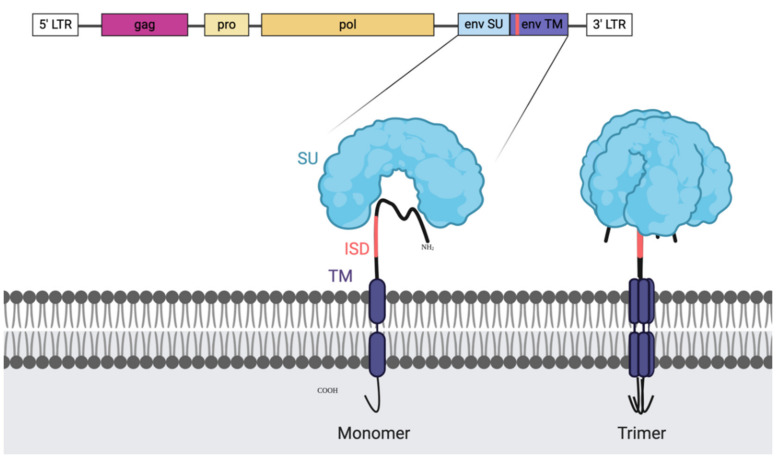
Illustration of endogenous retroviral genes and the envelope (ENV) protein with the immune suppressive domain (ISD). The retroviral genome organisation is shown on the top with the genes enclosed by the long terminal repeats (LTRs). First, in pink, we can find the group specific antigen (*GAG)*, next to it, in light yellow, is the protease (*PRO*) and after the polymerase (*POL*). Highlighted is the *ENV* gene and the generated protein as a monomer (left) and as the natural trimer (right). The ISD domain is recognisable in coral and is a part of the transmembrane domain (TM). It is partially shielded in the folded trimer by the surface unit (SU) which makes it a difficult, but not unreachable target and it is present on most retroviral ENV structures.

**Figure 2 ijms-21-04843-f002:**
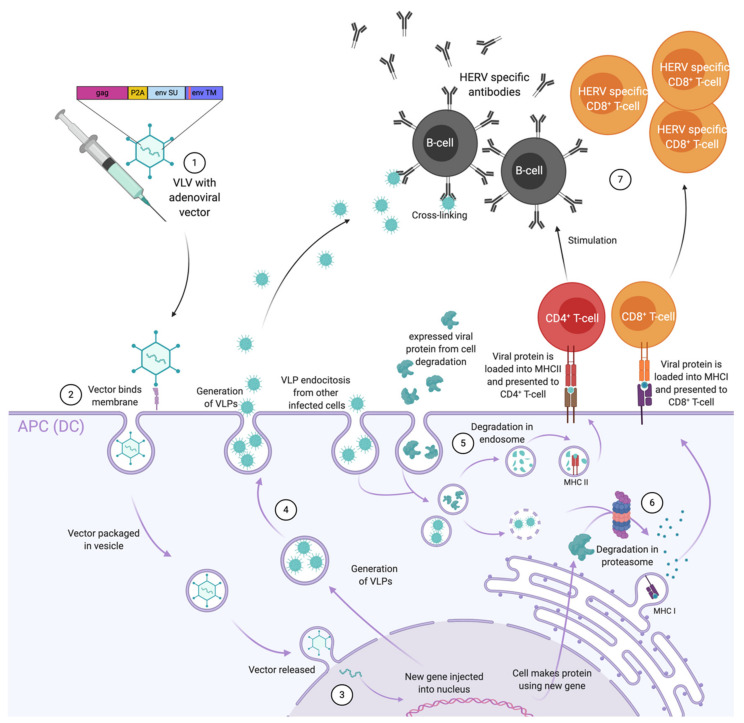
Illustration of the immune responses elicited by adenovirus based virus-like-vaccine (VLV) vaccination encoding endogenous retroviral (ERV) genes. **(1)** Vaccination with the adenoviral vector (Ad) encoding *GAG* and *ENV* genes, ideally harbouring mutations in the immunosuppressive domain (ISD) domain of *ENV*, is injected. **(2)** At the site of injection Ad directly infects professional antigen presenting cells (APCs) and releases the transgene into the recipient cell nucleus. **(3)** In the nucleus, the viral DNA codes for both viral and transgene proteins. Following their production, the fate of these proteins can be: **(4)** release of virus-like-particles (VLP)s to stimulate B-cells in an antigen structure dependent way; **(5)** uptake by APCs for endosomal degradation, presentation on major histocompatibility complex class II molecules (MHC-II), or **(6)** degradation in the proteasome (directly or after uptake) for presentation on major histocompatibility complex class I molecules (MHC-I) **(7)** stimulation of CD4^+^ T-cells and subsequent B-cell stimulation, and stimulation of CD8^+^ T-cells.

**Figure 3 ijms-21-04843-f003:**
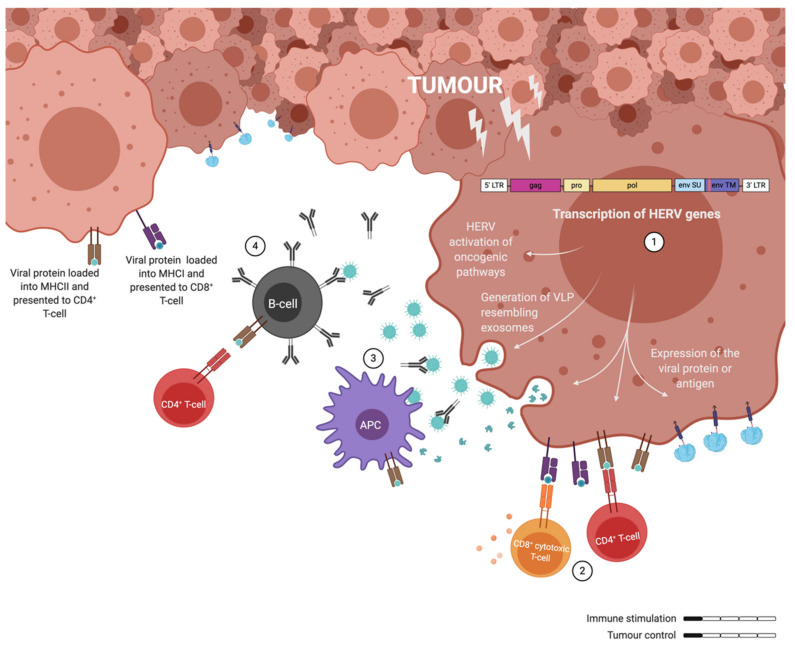
Illustration of the immune responses elicited by human endogenous retrovirus (HERV) expressing tumour cells. **(1)** ERV genes are transcribed in cancer cells and not only contribute to oncogenic pathways, but also to the release of exosomes (VLPs) and the presentation of antigens on the surface of cancer cells. **(2)** CD4^+^ and CD8^+^ T-cells can recognise ERV antigens, but they fail to get efficiently activated. **(3)** Similarly, antigen presenting cells (APCs) process the ERV particles and fail to elicit a potent T-cell response upon presentation. **(4)** Some activated B-cells can generate ERV specific antibodies. Overall, cancer associated retroviruses elicit non-protective immune responses through weak immune stimulation.

**Figure 4 ijms-21-04843-f004:**
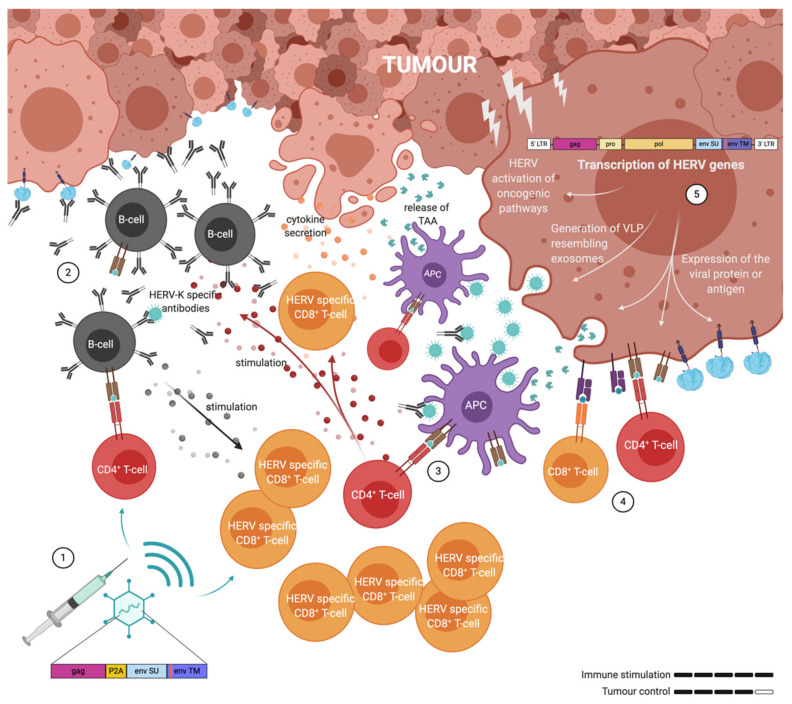
Illustration of the immune response elicited by adenovirus associated virus-like-vaccine (VLV) vaccination against human endogenous retrovirus (HERV) antigens presented and released by the tumour cells. **(1)** Immunisation has given rise to ERV specific B-cells, CD4^+^ T-cells and CD8^+^ T-cells. **(2)** B-cell derived antibodies bind cancer cells in order to block ERV immune suppression and facilitate antigen uptake and presentation on the surface of antigen presenting cells (APCs). **(3)** Vaccine induced CD4^+^ T-cells are activated by the antigen-MHCII complex presented on the surface of APC’s and stimulate other tumour infiltrating immune cells: B-cells and CD8^+^ T-cells. **(4)** Both CD4^+^ and CD8^+^ T-cells can directly contribute to tumour cell killing. **(5)** ERV genes are transcribed in cancer cells and not only contribute to oncogenesis, but also in the release of exosomes and the presentation of antigens on the surface of cancer cells.

**Table 1 ijms-21-04843-t001:** Overview table of the HERVs detected in several cancers^1^ [7,8,9,10,11,12,13,14,15,16,17,18,19,20,21,22,23,24,25,26,27,28,29,30,31,32,33,34,35,36,37,38,39,40,41,42,43,44,45,46,47,48,49,50,51,52,53,54,55,56,57,58,59].

	HERV-K	HERV-E	HERV-W	HERV-H	HEMO	HERV-FRD	HERV-R	HERV-P
Breast	X		X	X	X		X	X
Lymphoma	X		X	X				
Leukaemia	X						X	
Endometrial	X	X	X			X	X	
Prostate	X							
Seminoma	X		X					
TCC			X					
Ovarian	X	X			X		X	
Melanoma	X							
Lung	X			X	X		X	X
Colon	X		X	X				X
Pancreas	X							
Sarcoma	X							
Urothelial/Renal	X	X	X	X	X		X	
HNSCC	X						X	

^1^ be aware that lack of X only means that there is no registry of expression of that HERV in that cancer, not that it is not present.

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
