# Peer review of "Cancer Associated Endogenous Retroviruses: Ideal Immune Targets for Adenovirus-Based Immunotherapy"

_ijms, 2020, doi:10.3390/ijms21144843_

Round 1

Reviewer 1 Report

Comments for Authors:

This manuscript, submitted by Bermejo et al., describes recent progress of research on the association of endogenous retroviruses (ERVs) in mammals with cancers. Some ERVs are known to be highly expressed in various cancer cells and involved in tumor proliferation and can be used as an ideal target for the adenovirus-based immunotherapy. The review is written very well and could be an important basis for research on the treatment of cancer. I have a few minor points that the authors may wish to revise.

1) Page 2, line 88; "murine endogenous retrovirus" --> "murine ERVs"

2) Page 3, line 103; " This is an overview table of the HERVs detected in several cancers " --> " Overview of the HERVs detected in several cancers "

3) Page 4, line 163; "reported to be present" --> "reported to be expressed"

4) Page 5, line 212; "wich" --> "which"

5) Page 10, line 396-398; "(1) ERV genes are transcribed in cancer cells and not only contribute to oncogenesis, but also in the release of exosomes (VLPs) and the presentation of antigen are presented on the surface of cancer cells."

   This sounds odd.

Author Response

1) Page 2, line 88; "murine endogenous retrovirus" --> "murine ERVs" as the abbreviation was already mentioned before

2) Page 3, line 104; " This is an overview table of the HERVs detected in several cancers " --> " Overview of the HERVs detected in several cancers" as the determinant is not necessary for the title of the table

3) Page 4, line 165; "reported to be present" --> "reported to be expressed" as it is a more accurate description

4) Page 5, line 215; "wich" --> "which" as it is a typo

5) Page 10, line 399-401; "(1) ERV genes are transcribed in cancer cells and not only contribute to oncogenesis, but also in the release of exosomes (VLPs) and the presentation of antigen are presented on the surface of cancer cells." --> "(1) ERV genes are transcribed in cancer cells and not only contribute to oncogenic pathways, but also to the release of exosomes (VLPs) and the presentation of antigens on the surface of cancer cells." as the sentence was not well constructed and the information was not understandable.

Reviewer 2 Report

The manuscript by Bermejo et al. reviews cancer associated endogenous retroviruses (ERV) and as targets for adenovirus-based immunotherapy. The authors overview ERV in cancer, immune evasion, as targets in cancer, adenoviral vectors used as anti-retroviral vaccines, and adenovirus as therapeutic cancer vaccines with ERVs as targets. The following points need to be addressed.

Where is the Section 7, since there are section 6 and 8?

In figures 1, 3, and 4, what pro stands for? This needs to be provided in the abbreviation list.

References should be in uniform format, like not all have doi, such ref 120.

Author Response

1)Page 11, line 463: Section 8 --> Section 7 as the numeration was wrong and there is no Section 8

2)Page 13, line 509. pro: Protease abbreviation has been added to the list

3) References have been reviewed and DOI references have been added in all the articles that have one.